# Targeting Lipid Metabolism in Cancer Stem Cells for Anticancer Treatment

**DOI:** 10.3390/ijms252011185

**Published:** 2024-10-17

**Authors:** Manish Kumar Singh, Sunhee Han, Sungsoo Kim, Insug Kang

**Affiliations:** 1Department of Biochemistry and Molecular Biology, School of Medicine, Kyung Hee University, Seoul 02447, Republic of Korea; manishbiochem@gmail.com (M.K.S.); sunheehan@khu.ac.kr (S.H.); 2Biomedical Science Institute, Kyung Hee University, Seoul 02447, Republic of Korea; 3Department of Biomedical Science, Graduate School, Kyung Hee University, Seoul 02447, Republic of Korea

**Keywords:** autophagy, cancer stem cell, drug resistance, lipids, stemness, tumor microenvironment

## Abstract

Cancer stem cells (CSCs), or tumor-initiating cells (TICs), are small subpopulations (0.0001–0.1%) of cancer cells that are crucial for cancer relapse and therapy resistance. The elimination of each CSC is essential for achieving long-term remission. Metabolic reprogramming, particularly lipids, has a significant impact on drug efficacy by influencing drug diffusion, altering membrane permeability, modifying mitochondrial function, and adjusting the lipid composition within CSCs. These changes contribute to the development of chemoresistance in various cancers. The intricate relationship between lipid metabolism and drug resistance in CSCs is an emerging area of research, as different lipid species play essential roles in multiple stages of autophagy. However, the link between autophagy and lipid metabolism in the context of CSC regulation remains unclear. Understanding the interplay between autophagy and lipid reprogramming in CSCs could lead to the development of new approaches for enhancing therapies and reducing tumorigenicity in these cells. In this review, we explore the latest findings on lipid metabolism in CSCs, including the role of key regulatory enzymes, inhibitors, and the contribution of autophagy in maintaining lipid homeostasis. These recent findings may provide critical insights for identifying novel pharmacological targets for effective anticancer treatment.

## 1. Introduction

Cancer is a multifactorial disease and ranked as the second cause of death in the United States in 2020 and 2021 [1]. The National Cancer Institute GDC Data Portal reports an increasing number of cases every year, with the highest cases being bone marrow and blood cancer, followed by lung cancer. While advanced-stage cancer offers bleak prospects for complete recovery and reduces survival rates, early detection, accurate diagnosis, and effective treatment are crucial for improving a patient’s quality of life and increasing survival rates. Despites advancements in treatment, recurrence and relapse rates remain high, largely due to tumor heterogeneity and therapy resistance. CSCs have been identified as significant contributors to drug resistance and tumor recurrence.

CSCs use alternative sources of energy and altered metabolism to thrive in the tumor microenvironment [2]. They utilize substrates, such as glutamine, serine, and fatty acids in addition to glucose, to accomplish the energy requirement [3]. Alterations in lipid metabolism have been linked to cancer development, particularly in obese individuals. Studies have shown de novo lipid synthesis facilitates resistance to oxidative stress, and targeting them makes cancer cells susceptible to chemotherapy [4].

Autophagy is a key cellular pathway for protein degradation and nutrient recycling to maintain cellular homeostasis. Induced autophagy has been reported in CSCs, enabling them to adapt to the tumor microenvironment. Targeting these pathways, including lipid metabolism, opens avenues for identifying new targets for cancer treatment and shedding light on potential therapeutic interventions.

## 2. Cancer Stem Cells (CSCs)

CSCs represent a distinct and specialized subset of cells, typically comprising less than 1% of the total cancer cell population within tumors. These cells exhibit stem cell properties, including an enhanced capacity for self-renewal and the ability to differentiate into various tumorigenic cell types. CSCs are notably resistant to conventional therapies and play a pivotal role in tumor initiation, progression, and metastasis [5]. In 1997, John and Bonnet first identified cells with significant proliferative properties in acute myeloid leukemia (AML). They successfully isolated CSCs marked by the surface marker CD34^+^ and CD38^−^ [6]. The identification of these leukemia stem cells support the theory that CSCs, with self-renewing potential, derive tumorigenesis, akin to normal stem cells but with a pathological role in tumor progression [7]. Unlike normal stem cells, which possess a spectrum of differentiation potentials ranging from totipotency to unipotency [8], CSCs are integral to the malignant process, promoting the growth and spread of cancer. An enhanced expression of cell surface markers, such as CD44, CD36, CD90, CD133, and partner of SLD five 1 (PSF1) [9], transition into a dormant or quiescent state support stemness and drug resistance in CSCs (Figure 1) [10]. Hypoxia conditions and activation of specific genes related to epithelial-to-mesenchymal cell transition (EMT) strengthen the metastasis [11]. EMT is crucial for cancer cell metastasis and resistance to anoikis, a type of programmed cell death triggered by the detachment of epithelial cells. CSCs recruit immune cells, such as tumor-associated neutrophils (TAN), tumor-associated macrophages (TAM), and myeloid-derived suppressor cells (MDSC), that facilitate immunosuppression and support metastasis [12]. Various immune molecules, including IL-6, IL-4, IL-8, TGF-β, and TNF-α, support CSC proliferation and growth, aiding in CSC stemness in various cancers, such as breast, liver, oral squamous, and lung cancer (Figure 1) [13,14].

CSCs are regulated by various signaling pathways that are crucial for maintaining stemness and contribute to tumor progression. Peroxisome proliferator-activated receptors (PPARs), nuclear receptors involved in fat and glucose metabolism, play a critical role in modulating CSC characteristics. PPARs have three subtypes: PPARα, PPARδ, and PPARγ. In liver CSCs, activation of the PPARα pathway and enrichment of its downstream effector, stearoyl-CoA desaturase 1 (SCD1), is crucial for maintaining stemness [15]. The PPAR pathway also influences CSC traits through glucose metabolism. For instance, low PPARα expression in AML CSCs is inversely correlated with stemness. PPARα inhibits glucose metabolism by binding to HIF1α, reducing the expression of its downstream phosphoglycerate kinase 1 (PGK1) gene, thereby suppressing CSC stemness [16]. In hepatic CSCs, fatty acid 4-phenylbutyric acid (4-PBA) upregulates PPARα expression, preventing its degradation and promoting CSC initiation and tumorigenicity [17]. N1-methyladenosine methylation enhances PPARδ expression in hepatic CSCs, activating the PPAR pathway to regulate cholesterol metabolism, sustain stemness, and increase tumorigenicity [18]. PPARδ activation is also linked to high-fat, diet-induced colorectal cancer liver metastasis via upregulation of Nanog (Figure 2) [19]. Chen et al. reported that Nanog, a key regulator of stem cell fate, promotes the generation of stem-like tumor-initiating cells (TICs) and drives hepatocellular carcinoma (HCC) oncogenesis through metabolic reprogramming, specifically by shifting from oxidative phosphorylation (OXPHOS) to FAO [20]. Conversely, PPARγ activation inhibits migration in glioma stem cells (GSCs), reduces stemness in brain CSCs by downregulating SOX2 and YAP1, and promotes differentiation by inducing COL2A1 and HLXB9 expression [21,22].

Wnt/β-Catenin signaling is implicated in various cancers, including lung, liver, thyroid, colorectal, cervical, and glioblastoma. In the canonical Wnt signaling pathway, Wnt ligands bind to frizzled transmembrane receptors, activating disheveled, which interact with the T-cell factor/LEF family to stabilize and accumulate nuclear β-catenin, promoting transcriptional activity. During tumorigenesis, Wnt signaling enhances tumor migration and invasion by upregulating genes involved in cell adhesion, such as Eph/ephrin, E-cadherin, and matrix metalloproteinases (MMPs), thereby supporting stemness in CSCs [23]. In colorectal cancer cells, this pathway strongly derives tumor-initiating effects in CD44+ and CD133+ CSCs [24]. Farnesyl Dimethyl Chromanol (FDMC), a Wnt/β-Catenin pathway inhibitor, reduces stemness and metastatic potential in colorectal CSCs via inducing apoptosis [25]. Upregulation of Disheveled3 (DVL3) activates the Wnt/β-Catenin/c-Myc/SOX2 axis, enhancing CSC stemness and metastatic in colorectal cancer [26]. In gastric cancer, the ST2^+^ marker promotes metastasis via Wnt pathway activation and interaction with BCL-XL [27]. Similarly, in pancreatic cancer, elevated expression of Frizzled-7 (FZD7) derives the CSC phenotype and liver metastasis through the canonical Wnt/β-Catenin pathway (Figure 2) [28].

Hedgehog (Hh) has been implicated in the tumor-initiating function of CSCs. In hepatic CSCs, the Hh pathway is maintained by the caspase-3/Sterol-regulatory element-binding Protein-2 (SREBP2) axis, which supports cholesterol synthesis, stemness, and tumorigenicity. TRNA methyltransferase 6 (TRMT6)/TRMT61A-mediated N1-methyadenosine methylation in liver CSCs promotes cholesterol metabolism and activates the Hh pathway. Maintaining stemness also enhances tumorigenicity (Figure 2). Hh signaling is further involved in maintaining the breast CSCs through the stabilization of GLI1 transcription [29].

The Notch pathway plays a significant role in CSC metastasis in various cancers, including breast, glioma, renal, and ovarian cancers. In breast cancer, Bone Morphogenetic Protein 4 (BMP-4) enhances stemness and EMT programs by activating the Notch pathway in a Smad4-dependent manner [30]. Upregulated Notch signaling in renal CSCs contributes to malignant behaviors, such as metastasis, stemness maintenance, and tumorigenesis [31]. Continuous activation of the Notch pathway leads to overexpression of CSC marker genes, resulting in enhanced tumorigenesis compared to cells with normal Notch activity [32].

The NF-κB pathway is crucial for maintaining CSC stemness and aggressive metastasis. In glioma stem cells (GSCs), NF-κB activation supports the proneuronal to mesenchymal transition (PMT) [33]. In CD133^+^ liver CSCs, BMI-1-induced NF-κB activation and nuclear translocation promote CSC stemness and metastatic potential while inhibiting apoptosis [34]. Ovarian CSCs exhibit high expression of NF-κB pathway-related proteins, and inhibiting the NF-κB reduces the CSC population [35]. In breast cancer, elevated expression of stromal cell-derived factor-1 (SDF-1) induces stemness and EMT by activating the NF-κB pathway (Figure 2) [36]. In colon cancer inhibition of NF-κB, using dimethylamine parthenolide (DMAPT) altered the metabolomic profiles, reduced unsaturated lipid levels, and impaired spheroid formation in colon CSCs, underscoring this pathway in preserving CSC stemness [37,38].

The JAK/STAT pathway is involved in maintaining CSC stemness across various cancers. In breast cancer, inhibiting the JAK/STATE reduces the expression of stemness-related genes [39]. Activation of the lipid metabolism-related STAT3/CPT1B/fatty acid β-oxidation (FAO) axis in breast CSCs is positively correlated with stemness maintenance [40]. Immunity plays a crucial role in activating the JAK/STAT pathway in CSCs, as Il-6 secreted by regulatory T-cells induces the STAT3 pathway in glioma cells, sustaining the stemness phenotype (Figure 2) [41]. In prostate CSCs, IL-6 promotes tumorigenesis, and its inhibition eliminates tumor initiation. The interferon-inducible (IFN-inducible) protein viperin overexpression in CSCs partially inhibits FAO through the JAK/STAT pathway, reprogramming metabolism to activate tumor progression. However, viperin-expressing cells have more accumulation of LDs, compared to cells with basal level of expression [42].

The TGF-β pathway is closely linked to CSC tumorigenesis and stemness. In various cancers, including ovarian and oral squamous carcinoma, inhibiting the TGF-β pathway suppresses CSC stemness and EMT [43,44]. Paired-Related Homeobox 1 (Prrx1) activates the TGF-β/Smad pathway in GSCs, inducing stemness and promoting vascularization and therapeutic resistance (Figure 2) [45].

The PI3K/AKT pathway is pivotal in driving the differentiation of normal stem cells into CSCs in various cancers [46]. In breast cancer, continuous activation of PI3K/AKT pathway by PD-L1 is crucial for maintaining CSC stemness (Figure 2) [47]. In breast CSCs, the transmembrane and coiled-coil domain family 3 (TMCC3) activates PI3K/AKT via binding to AKT, promoting tumorigenesis and metastasis (Figure 2) [48]. In liver cancer, the tumor suppressor Connexin 32 (Cx32) attenuates PI3K/AKT pathway activity, suppressing stemness and tumorigenicity [49]. Stress-induced phosphoprotein 1 (STIP1) enhances MMP2 and MMP-9 expression in osteosarcoma by activating the PI3K/AKT and ERK1/2 pathways, ultimately promoting CSC metastasis [50].

CSCs undergo metabolic reprogramming to sustain proliferation and survival, particularly under nutritional deficiency. They primarily rely on aerobic glycolysis (Warburg effect) [51] and adjust their metabolism to meet their energy requirements [52]. Reprogrammed lipid metabolism, including lipid uptake, de novo lipogenesis, fatty acid activation, desaturation, chain elongation, FAO, and lipid droplet (LD) storage and degradation, supports cancer growth and proliferation [53], enabling CSCs to evade cell death and promotes cancer progression [54]. Mitochondrial metabolism is vital for fulfilling the energy demands necessary for sustained CSC proliferation while also supplying substrate required by various cellular compartments. Studies have shown that poorly differentiated counterparts demonstrate increased glycolytic activity, primarily due to impaired differentiation in CSCs [55]. In breast CSCs, the overexpression of pyruvate dehydrogenase kinase 1 (PDK1) suppresses mitochondrial glycolysis. The depletion of PDK1 levels significantly decreased the population of ALDH1-positive BCSCs, hindering their ability to form spheroids [55].

Additionally, CSCs display high levels of unsaturated fatty acids, and inhibiting key enzymes, such as stearoyl-COA desaturase 1 (SCD1) and acetaldehyde dehydrogenase 1 A1 (ALDH1A1), prevents apoptosis and enhances stemness [56]. Elevated ALDH1 levels are linked to various cancers, including lung cancer [57], breast cancer [58], esophageal cancer [59], and colon cancer [60], where it contributes to enhance their stemness and tumorigenicity. ALDH1A1 inhibitor (CM037) significantly decreased levels of unsaturated lipids and suppressed spheroid forming of CSCs [38]. SCD1 inhibitor, CAY10566, effectively reduced unsaturated lipid levels in CSCs and inhibited spheroid formation, suggesting that SCD1 activity is essential for CSC stemness and tumorigenicity [38]. These findings suggest that glycolytic intermediates may be redirected to fuel increased lipogenesis in CSCs, further enhancing their stemness. Therefore, targeting intermediates within the glycolytic pathway may provide a promising strategy to eliminate CSCs, reducing tumor growth, metastasis, and recurrence in cancer patients [61].

In ovarian cancer, CSCs prefer fatty acid β-oxidation over the Warburg effect [62]. Glioblastoma CSCs (GBM-CSCs) express high levels of fatty acid synthase (FASN), and inhibiting FASN reduces their proliferation and migration [63]. In head and neck squamous cell carcinoma, ALDH^+^ cells inhibit T-cell proliferation and secrete cytokines, such as interferon γ (IFN-γ), IL-2, and tumor necrosis factor-α (TNF-α), promoting tumorigenesis and drug resistance [64]. Orlistat has been shown to inhibit FASN activity in CRC cell lines (CaCo-2 and SW480) [65]. In vivo studies with HT-29/tk-luc human colorectal carcinoma-bearing animal models have demonstrated Orlistat’s capacity to suppress tumor growth [66]. Additionally, novel FASN inhibitors, TVB-3166 and TVB-3664, currently in Phase II clinical trials, significantly reduced tumor volume in patient-derived CRC xenograft models [67]. Natural compounds like (E)-N-(2-(4-methoxystyryl) phenyl) furan-2-carboxamide have also shown promising chemo-preventive activity in HCT116 cells [68]. Further, investigation of the combined effects of ALDH and FASN inhibitors, as well as their derivatives, across various cancers is crucial to explore their potential in preventing cancer progression by inhibiting CSC growth and progression.

## 3. Lipid Metabolism in CSCs

Lipid metabolism has been associated with several metabolic diseases and pathological cancers. Recent evidence has shown lipid metabolism is crucial for maintenance of CSCs properties. CSCs exhibit elevated LDs compared to normal cells. During metabolic stress resulting from inhibiting glycolysis, free fatty acids (FFAs) from LDs are used for ATP production through FAO. Dysregulation of lipids poses a significant risk for cancer development, particularly in a condition known as cachexia, characterized by severe weight loss, physical deterioration, loss of muscles and adipose tissue, anorexia, inflammation, and insulin resistance [69]. Lipolysis in adipose tissue is associated with tumor progression by supplying energy to cancer cells [70]. For instance, increased levels of circulating free fatty acids, mono-acylglycerides, and di-acylglycerides have been observed in cachectic ovarian cancer patients [71].

Increased uptake of lipids contributes to the accumulation of LDs, thereby enhancing the tumor-initiating capacity of CSCs [72]. Elevated lipid content in colorectal cancer stem cells (CRCSCs) has been associated with greater clonogenicity and tumorigenicity [73]. Notably, the inhibition of phospholipase A2 leads to a reduction in LDs and triggers apoptosis [74].

Regulation of the transport of long-chain fatty acids into the mitochondria for FAO is mediated by changes in the expression of carnitine palmitoyl transferase 1A (CPT1A) (Table 1). In colorectal CSCs, increased LDs serve as unique markers and are induced by hypoxia via HIF-1 and HIF-2 mediated repression of CPT1A [75]. Increased CPT1A expression promotes tumor progression and metastasis by inhibiting cell death (apoptosis) or detachment-induced cell death (anoikis) in breast cancer and leukemia [76]. CPT1A is involved in a positive feedback loop with cMyc. For instance, CPT1A-cMyc loop predominantly suppresses tumoral ferroptosis via NRF2/GPX4 and CSL4/PUFA-PLs pathways in lung cancer [77]. However, the molecular interaction of c-Myc and CPT1A in FAO and PUFA-PLs loop is poorly understood. Thus, the amount of various lipid species and fatty acids, which are basic constituents of lipid membranes and key players in signal transduction pathways, has revealed a significant role in the maintenance of CSC stemness [78].

CSCs are characterized by a high content of unsaturated lipids, including monounsaturated fatty acids (MUFAs), which have been shown to influence CSC characteristics [79]. The enzyme SCD1 facilitates desaturation and is critical for CSC generation and maintenance in various cancers, such as ovarian, breast, and liver cancers [37,80]. In colorectal CSCs, SCD1 and ALDH1A1 significantly contribute to maintaining stemness [38]. Inhibition of desaturases in ovarian cancer reduces cancer stemness, thereby increasing their susceptibility to treatment [37]. Sterol regulatory element binding proteins (SREBPs), a family of helix-loop-helix leucine zipper transcription factors, play a pivotal role in the regulation of lipogenesis. They induce the biosynthesis of FAs and cholesterol while regulating several key lipogenic enzymes such, as ATP citrate lyase (ACLY), ACC1, and FASN, all of which support CSC stemness (Table 1) [81]. Modification in SREBP1 has been linked to increased lipogenesis, tumor proliferation, and poor prognosis in hepatocellular carcinoma (HCC) [82]. Lipid profiling in osteosarcoma patients has revealed elevated levels of cholesterol, choline, polyunsaturated fatty acids (PUFAs), and glycerol during metastasis [83]. However, the precise mechanism by which desaturase inhibitors affect CSCs remains to be fully elucidated. In this context, metabolomic and lipidomic approaches could prove valuable in studying different stages of cancer, with lipid profiling offering the potential for identifying biomarkers, monitoring cancer progression, aiding in diagnosis, and novel therapeutic targets in various cancers [83].

**Table 1 ijms-25-11185-t001:** List of Enzymes Involved in Lipid Metabolism and Dysfunction in Various Cancers.

Enzymes	Molecular and Cellular Activity	Effects in Cancer	References
Acyl-CoA synthase long-chain 3 (ACSL3)	Converts free fatty acids into fatty acyl-CoA	In ER-negative breast cancer development and progression	[84]
Acetyl-CoA carboxylase1/2	Promotes conversion of acetyl-CoA to malonyl-CoA (associated with SCD)	Support tumor growth	[85]
Acyl-CoA cholesterol acyltransferase 1 (ACAT1)	Involved in cholesterol esterification Inhibits apoptosis via increase expression of caspase 3/7 activity Decreased mitochondrial membrane potential	Leukemia, glioma, breast, pancreatic, ovarian, and prostate cancer	[86]
Cyclooxygenase 2	Involved in inflammation and tumor stroma interaction	Promotes tumor growth	[87]
Choline kinase	Required for synthesis of Phospholipids	Poor prognosis for cancer	[88,89]
Carnitine palmitoyl-transferase-1A (CPT1A) and 1B	Formation of fatty acyl-carnitine and transport of fatty acyl-carnitine and long-chain fatty acids across the inner mitochondrial membrane	Breast cancer, prostate cancer, lung cancer, and ovarian cancer metastasis	[90]
Diacylglycerol (DAG)-acyltransferase, diacylglycerol O-acyltransferase 1(DGAT1) and DGAT2	Catalyzes the esterification of fatty acid (FA)-CoA with DAG to produce triglycerides, a main component of LDs	In glioblastoma, acts as an antioxidant and prevents cell death	[91]
2, 4,-dienoyl CoA reductase (DECR1/2)	FAO of PUFAs depends on NADPH	Castrate-resistant prostate cancer (CRPC)	[92]
Fatty acid desaturases	Facilitates the conversion of saturated to unsaturated fatty acids, regulated by NF-κB	Facilitates EMT conversion	[93]
Fatty acid 2-hydroxylase (FA2H)	Hydroxylation of the C-2 position of free fatty acids	Promotes lung metastasis and invasion	
3-hydroxy-3-methylglutaryl-CoA synthase (HMGCS1/2) and 3-hydroxy-3-methylglutaryl-CoA reductase (HMGCR)	Involved in cholesterol synthesis, Acetyl-CoA ketogenesis in mitochondria	In ER-breast cancer metastasis and gastric cancer	[86]
Lipoxygenase (LOXs), ALOX15	LOX5/12/15 catalyze lipid peroxidation, mainly in a Fe^2+^—dependent manner	In gastric cancer-promoting ferroptosis	[94]
Lysosomal acid lipase (LAL)	Controls neutral lipid metabolic signaling Hydrolyzes cholesteryl ester and triglycerides in lysosomes Produces FFA and cholesterol	Role in myeloid-derived suppressor cells and tumorigenesis	[95]
Phospholipase A2	Hydrolyze the sn-2 ester bond of the phospholipids	In ovarian cancer	[96]
Stearoyl-CoA desaturase (SCD1)	Introduces a double bond at Δ^9^ position in fatty acids Produces more MUFAs; Triggers an iron-dependent cell death called ferroptosis	Poor prognosis in Stage II colon cancer, lung cancer, and endometrial cancer	[97]
Spinster homologue 2 (SPNS2)	Transporter for lipid sphingosine-1-phosphate Helps in lymphocyte trafficking	Tumor progression, lung metastasis, hepatocellular carcinoma, and poor prognosis in cancer	[98]
Sterol regulatory element binding proteins (SREBPs)	Helix-loop-helix leucine zipper transcription factors Activates storage of cholesterol and fatty acids into LDs Transcriptional regulation of lipogenesis Poor prognosis in cancer	In HCC tumorigenesis and metastasis	[99]

## 4. Cholesterol Metabolism in CSC

The presence of elevated cholesterol levels has been linked to cancer progression and poor prognosis. They are synthesized via mevalonate (MVA) pathway, which is upregulated in various cancers, including breast, lung, and HCC. Targeting this pathway has shown to decrease cell proliferation. Cholesterol biosynthesis leads to the formation of lipid rafts, which sequester proteins such as CD44, CD24, CD133, and CXCR4 associated with CSCs [91]. The MVA pathway primarily regulates the biosynthesis of steroid hormones, cholesterol, and nonsteroidal isoprenoids. Targeting key enzymes of MVA pathway, such as 3-hydroxy-3-methylglutaryl-CoA synthase (HMGCS1) and 3-hydroxy-3-methylglutaryl-CoA reductase (HMGCR), reduces cancer cell proliferation and migration [69]. Carnitine palmitoyl-transferase 1 (CPT1) assists the transfer of long-chain fatty acids into the mitochondrial membrane for oxidation, promoting acetyl-CoA and ATP production, thus promoting CSC growth [92]. In CRC metformin, an AMPK activator, along with an HMG-CoA reductase inhibitor, or a mTOR inhibitor, significantly reduces the CSC population. However, the CSC population rebounded after the addition of mevalonic acid. This suggests that mevalonic acid mitigates the inhibitory effects of these therapies on CSCs [100].

Increased cholesterol uptake through low-density lipoprotein receptors (LDLR) is associated with poor prognosis in breast and lung cancer [93]. Scavenger receptor class B type I (SR-BI) facilitates cholesterol uptake via high-density lipoproteins (HDLs). Targeting SR-BI reduces cholesterol uptake and proliferation in breast and prostate cancers [94]. Another enzyme, acyl-CoA cholesterol acyltransferase 1 (ACAT1), assists in cholesterol esterification, which is essential for cancer cell growth. Inhibition of key enzymes such as ACAT1 by avasimibe reduces tumor growth and metastasis in vivo in prostate cancer. A similar inhibitor, avasimin, induces apoptosis in various cancers, including colon, pancreatic, lung, and prostate cancer, exhibiting promising anticancer effects [95]. Liver X receptor (LXR), a nuclear hormone receptor with an important function in cholesterol metabolism, consists of two isoforms: LXRα (NR1H3) and LXRβ (NR1H2), expressed in the liver, intestine, adipose tissue, and macrophages. The LXR signaling agonist GW3965 reduces LDR receptors and increases ABC transporter A1 (ABCA1) expression, resulting in decreased intracellular cholesterol and reduced proliferation in clear cell renal cell carcinoma [96]. Atorvastatin, a lipophilic statin, has shown increased survival in HMGCR-positive breast cancer. Another drug, such as medroxyprogesterone acetate (MPA), suppresses the breast tumor spheres via inhibition of cholesterol synthesis [101]. These findings indicate that cholesterol homeostasis is important for CSC growth and survival; thus, this could be a promising target in various cancers.

## 5. Fatty Acid Oxidation in CSCs

CSCs rely on increased FAO for growth and survival and assists in drug resistance. The FAO process in CSCs utilizes acetyl-CoA and NADH to produce ATP, maintaining energy homeostasis that contributes to their resilience against treatment. Higher levels of certain enzymes, such as ATP-citrate lyase (ACLY), ACSS1, and ACSS2, have been observed in certain cancer types, promoting tumor growth and metastasis by providing an alternative energy source to glucose (Table 1) [102]. For instance, FAO supports the maintenance of stemness in mesenchymal stem cells in advance gastric cancer [103]. In pancreatic ductal adenocarcinoma (PDAC), CSC and non-CSC-targeting OXPHOS significantly impair the tumorigenicity and chemoresistance. PaCSCs, enhanced expression of lipid metabolism genes such as MGLL, PPARD, and CPT1A, have been correlated with poor prognosis with worse outcomes. Moreover, CPT1A overexpression was significantly effective in circulating PaCSCs compared to primary tumors, suggesting a survival advantage for cells with increased lipid storage and metabolism in the blood. PaCSC-enriched conditions (spheres or CD133^+^ cells) showed a significantly high LD content than non-CSCs (adherent cells or CD133^−^), suggesting that the differential distribution of lipid content is dependent on stemness. Mascaraque et al. have demonstrated that inhibition of FAO enhanced the sensitivity of PDAC cells to chemotherapy drugs, such as gemcitabine [104]. Furthermore, FAO activates the oncogenic protein Src and promotes CSC metastasis in triple-negative breast cancer (TNBC) [105]. Metastatic TNBC maintains high levels of ATP through FAO and activates the Src oncoprotein via autophosphorylation at tyrosine 419 (Y419) [105]. Moreover, leukemic stem cells (LSCs), deficient in CPT1A, the rate-limiting enzyme in FAO, exhibit resistance to avocatin B, a lipid derived from avocado that selectively targets AML stem cells with minimal impact on normal counterparts [106]. These findings underscore the critical role of FAO in the development of chemoresistance in CSCs.

Lipid peroxidation of specific fatty acids has been identified as a trigger for ferroptosis, an iron-dependent form of cell death notably observed in RAS-mutated cancer cells [107]. The excessive iron-catalyzed peroxidation of membrane phospholipids, particularly those containing the polyunsaturated fatty acid arachidonic acid (AA), plays a critical role in driving ferroptosis [108]. Phadnis et al. reported that MMD, a scaffold protein located in the Golgi apparatus, increases susceptibility to ferroptosis in ovarian and renal cancer cells through a mechanism dependent on acyl-coenzyme A (CoA) synthetase long-chain family member 4 (ACSL4) and membrane-bound O-acyltransferase family member 7 (MBOAT7) [109]. This mechanism facilitates the incorporation of AA into phosphatidylinositol (PI), thereby enhancing ferroptosis sensitivity in OVCAR-8 ovarian cancer cells [110]. However, the biochemical mechanisms underlying ferroptosis in other cancer types remain to be thoroughly investigated and compared with non-cancerous cells, as well as in in vivo models.

The AMPK pathway is crucial for energy metabolism and regulates mitochondrial FAO. AMPK inhibits certain enzymes, such as ACC1 and ACC2, leading to increased FAO and NADPH production, supporting cancer cell survival [111,112]. The AMPK/mTOR pathways have been shown to regulate lipid metabolism through different mechanisms, with AMPK promoting FAO, while mTORC1 enhances lipogenesis [113]. AMPK inhibitors, such as metformin, dorsomorphin (Compound-C), and Bay-3827, have been found to decrease FAO, potentially inhibiting cancer cell survival, although their off-target effects are a significant concern [114]. The role of AMPK/mTOR pathway in maintaining lipid metabolism in CSCs under nutritional deficiency warrants further investigation [115].

## 6. Signaling Pathways Involved in Lipid Metabolism in CSCs

Lipid reprogramming in CSCs assists self-renewal, proliferation, and resistance to treatment [116]. Alterations in lipid metabolism lead to enhanced lipogenesis, increased free lipids content in LDs that can be easily transferred to other cells [117]. The Pi3K/AKT/mTOR, JAK/STAT, Hippo, Wnt, and NF-κB signaling pathways are essential for regulating lipid metabolism in CSCs. Activated PI3K/AKT pathway supports cell growth through increased PIP3 production in various cancers (Figure 3) [118]. In prostate cancer, the silencing of the tumor suppressor PTEN enhances the expression of enzymes involved in FA synthesis, promoting CSC metastasis and growth [119].

Notch signaling is implicated in lipid metabolism. Notch1 signaling can regulate the expression of peroxisome proliferator-activated receptor α (PPARα) and lipid oxidation genes to maintain lipid levels in hepatocyte and adipocyte cells [120]. The selective elimination of colon CSCs can be achieved by targeting Notch and Wnt signaling, specifically inhibiting SCD1 (Figure 3) [121]. Notch 1 plays a key role in utilizing FAO to maintain redox balance in quiescent endothelial cells (QECs). Supplementation with acetate, which is metabolized to acetyl-CoA, restores endothelial quiescence and counteracts oxidative stress-induced endothelial dysfunction in CPT1A-deficient mice, highlighting potential therapeutic avenues. Thus, QECs rely on FAO for vasculoprotection against oxidative stress [122].

The Hippo-YAP/TAZ signaling pathway regulates the stemness in CSCs. In breast cancer, the Hippo pathway facilitates geranylgeranylation (GGylation)-dependent cell proliferation and migration [123,124]; however, the YAP signaling pathway promotes cell proliferation, migration, and invasion in gastric cancer [124]. The YAP or TAZ also facilitates self-renewal and tumorigenesis in CSCs [125,126,127]. In lung cancer, SCD1 plays a role in maintaining fatty acid metabolism and promotes CSC stemness by stabilizing YAP/TAZ and promoting their nuclear localization (Figure 3) [128]. Further, silencing of SCD1 induced selective apoptosis of ALDH-1A1-positive cells and impaired in vivo tumorigenicity of 3D lung CSCs [129].

The JAK/STAT pathway, commonly activated in many cancers, plays a role in hematopoiesis and self-renewal of normal embryonic stem cells. Inhibition of JAK/STAT signaling reduces CSC stemness in AML [130]. Targeting JAK/STAT3 pathway via CPT1B inhibitors blocks breast CSC self-renewal and tumor growth in vivo [40].

Wnt signaling is implicated in regulating lipogenesis in CSCs [131]. In CSCs, the β-catenin pathway regulates fatty acid metabolism through YAP/TAZ signaling, particularly by modulating SCD1. In vivo studies have shown that the Wnt/β-catenin pathway is linked to lipid metabolism in mouse liver hematopoietic stem cells, where SCD-derived MUFAs establish a positive feedback loop by stabilizing and upregulating Lrp5/6 mRNA, thus enhancing Wnt signaling (Figure 3) [128]. MUFAs are essential for the synthesis and release of Wnt ligands. In CRC, Wnt-signaling dysregulation and a high-fat diet disrupt bile acid distribution, activate FXR, and drive the malignant transformation of the Lgr5+ CSC subpopulation, promoting the progression from adenomas to adenocarcinomas [132]. Additionally, the Wnt/β-catenin pathway regulates lipogenesis and MUFA production, as well as de novo adipogenesis in BCS through the upregulation of ACC, FASN, and SREBP1-c expression [133].

Moreover, various antioxidant pathways, such as Glutathione peroxidase 4 (GPX4), ferroptosis inhibitory protein 1 (FSP1), dihydroorotate dehydrogenase (DHODH), Fas-associated factor 1 (FAF1), and Tetrahydrobiopterin (BH4), act as antagonists by inhibiting ferroptosis under ROS-induced stress (Figure 3) [134,135,136,137]. The nuclear factor erythroid 2 (Nrf2), a key transcription factor in cellular antioxidant responses, upregulates ferroptosis-inhibiting molecules to prevent ferroptosis [138].

The Hh signaling pathway plays a crucial role in maintaining CSC stemness, particularly by influencing EMT in breast cancer and mammary gland morphogenesis [139]. It supports the expansion of mammary progenitors derived from mammary stem cells and regulates the differential usage of the TP63 promoter, thereby enhancing clonogenicity [140]. Cyclopamine, an inhibitor of the transmembrane protein SMO, has shown efficacy against CML stem cells in vitro and in vivo [141], while GANT61, a GLI, inhibitor, blocks GLI function, inhibiting tumor cell growth (Figure 3). GANT61 has demonstrated potent effects in inducing cell death in colon carcinoma, neuroblastoma, and chronic lymphocytic leukemia [141,142,143]. GLI1 expression is upregulated in CD34+ subpopulation of AML cells. GANT61 induces AML cells’ apoptosis and differentiation. The combination of GANT61 with chemotherapeutics has shown a synergistic anti-proliferative effect on primary CD34+ AML cells [144]. These findings highlight the intricate connection between Hh signaling, lipid metabolism, and CSC regulation, suggesting potential therapeutic targets for cancer treatment.

## 7. Autophagy as an Essential Player in Maintaining CSC Stemness

Autophagy is a lysosomal-dependent mechanism crucial for metabolic recycling and cell survival under conditions of stress and nutrient deprivation [145]. This process involves various steps: initiation, autophagosome formation, elongation, and, finally, lysosomal degradation [146]. Autophagy plays a vital role in maintaining intracellular homeostasis by removing damaged organelles, misfolded or damaged proteins, recycling them to sustain cellular homeostasis. This mechanism is intricately associated with the preservation of CSC stemness and consequently facilitates tumorigenesis and drug resistance [147,148]. CSCs often exhibit elevated levels of autophagy, as indicated by the increased expression of autophagy-related genes, such as ATG4, ATG5, and Becline1, displaying an elevated autophagic flux (Figure 4) [149]. Interestingly, while autophagy supports tumor growth by maintaining CSC viability, it also acts as a tumor suppressor by destabilizing the transcription factor NRF2, thus helping tumor cells resist oxidative stress [150].

The role of autophagy in CSC proliferation and tumorigenesis is context-dependent, involving processes such as lipophagy, where LDs merge with autophagosomes, and increased FAO, which further amplifies the growth-promoting effects in CSCs [151]. Forkhead box 3A (FOXA3), a transcription factor, induces the expression of autophagy-related genes, such as LC3, ULK1, ATG5, and GABARAPL1 and beclin-1 in stem cells, and has been implicated in regulating CSC fate (Figure 4). The loss of FOX3A has been shown to enhance CSC stemness in various cancers, such as glioblastoma, ovarian, breast, liver, and colorectal [152,153,154]. Moreover, the knockdown of ATG5 in ovarian cancer stem cells reduces chemoresistance and self-renewal capacity (Figure 4) [155]. Moreover, the basal level of autophagy/mitophagy is higher in BCSCs compared to normal tissue stem cells, and autophagy induced the upregulation of CD44 and vimentin, both of which are recognized as stem cell markers [156].

Mitophagy, a specialized form of autophagy that targets damaged mitochondria for degradation, is implicated in CSC maintenance. Elevated ROS levels can impair normal mitochondrial functions, leading to cell death. However, CSCs possess enhanced anti-oxidative capacity, which protects them from ROS-mediated cell death. In hematopoietic stem cells (HSCs), the knockdown of PINK1/parkin suppresses mitophagy and reduces stemness [157]. CSCs leverage elevated mitophagy to regulate ROS and FAO, thus supporting their survival and proliferation [158].

Autophagy inhibitors have shown efficacy in sensitizing resistant tumor cells to chemotherapy and radiotherapy. For instance, drugs such as clomipramine and chloroquine, have been demonstrated to induce cell death in enzalutamide (ENZA)-resistant prostate cancer cells [159]. The combination of standard cancer therapies with autophagy inhibitors has proven to be more effective than either approach alone in targeting CSCs [160]. In GSCs, combining paclitaxel (PTX) and chloroquine phosphate (CQ) enhanced the efficacy of chemotherapy [161]. Furthermore, combining autophagy inhibitors like CQ and hydroxychloroquine (HCQ) with other cancer drugs has been shown to improve treatment outcomes, as evidenced in gastrointestinal stromal tumors where the combination of CQ and imatinib promotes apoptosis [149,162].

## 8. Lipid Metabolism and Autophagy Pathway Crosstalk in CSC

The metabolic reprogramming within TME is influenced by multiple factors, including enhanced FAO, heightened lipid uptake, and elevated expression of FASN. These alterations result in higher levels of lipids and fatty acids (FAs), leading to the induced accumulation of LDs in cancer cells. In CSCs, FAs are stored in LDs as triacylglycerols (TAGs) and sterol esters, maintaining lipid homeostasis and preventing lipotoxicity [54]. Autophagy plays a pivotal role in regulating the amount of lipids and FAs in CSCs by breaking down LDs and releasing FAs through lipophagy (Figure 4). This phenomenon has been observed in various cancers and can suppress tumor growth through mechanisms such as ferroptosis, ROS production, and ER stress [163].

Autophagy also supports CSC stemness by providing essential nutrients, particularly under stress conditions, such as hypoxia, nutrient deprivation, energy ablation, and exposure to chemotherapy or radiation [115]. Under starvation conditions, TAGs stored in cytoplasmic LDs are degraded through lysosomes-mediated autophagy, a process driven by patatin-like phospholipase domain-containing protein 2 (PNPLA2). This breakdown releases free FAs, which can either be transported to the mitochondria for β-oxidation or re-esterified into LDs [164]. Additionally, a deficiency in glucose-6-phosphatase (G6P), whether induced genetically or chemically, has been shown to trigger autophagy, reducing lipid accumulation and liver steatosis. This suggests that G6P could be a promising target for reducing lipid buildup and limiting tumorigenesis in cancer cells [165].

Lysosomal acid lipase (LAL) is essential for autophagy and lipid degradation. Inhibition of LAL has been linked to hematopoietic abnormalities and the suppression of immature myeloid-derived suppressor cells (MDSCs), which can impair immune surveillance and promote tumorigenesis [95,166].

## 9. Targeting Lipid Metabolism and Autophagy Pathway in CSCs

Cancer therapies targeting the TME through lipid metabolism, FAO, and degradation pathways, such as lipophagy and ferroptosis, offer promising approaches to inhibit tumorigenesis (Figure 5). Modulating these pathways can alter lipid uptake in cancer cells, including CSCs, significantly affecting cancer progression. Inhibition of key enzymes like ACC and FASN has shown considerable efficacy in eliminating CSCs. Targeting ACC in pancreatic cancer cells has been shown to effectively inhibit tumor proliferation both in vivo and in vitro, which is mediated through the reduction of palmitoylation key ligands involved in the Wnt and Hh signaling pathways, which are critical for maintaining CSC characteristics [167]. Inhibiting FASN in HER2-positive advanced breast cancer cells has reduced cancer growth in both preclinical and clinical studies. Soraphen A, an inhibitor, has shown a potential effect on breast CSCs and has proven effective in NSCLC and HCC cancers (Table 2) [61,168,169]. Targeting CPT1A, a critical enzyme in FAO, with inhibitors such as etomoxir and ranolazine, has been shown to reduce fatty acid transport into mitochondria, thereby suppressing tumor growth [170]. Napabucasin (BBI608) has demonstrated significant efficacy in metastatic colorectal and pancreatic cancer [171]. Additionally, targeting ALDH+ breast CSCs with disulfiram and gemcitabine has led to tumor growth inhibition and enhanced T-cell mediated immunity [172]. Fresolimumab, a TGF-β blocker, has improved immune response and extended survival in clinical studies involving breast cancer patients [173]. ALDH inhibitors, such as disulfiram and its derivatives, have also enhanced chemotherapy sensitivity in lung cancer [174].

Breast CSCs exhibit increased levels of long-chain FAO metabolites, and treatment with etomoxir has successfully inhibited FAO, reducing stem cell viability and tumor sphere formation [40]. Furthermore, inhibitors targeting the Hh signaling pathway, such as Sonidegib (LDE225) and Glasdegib (PF-04449913), have shown long-term survival benefits in patients with basal cell carcinoma (BCC) and acute myeloid leukemia (AML), respectively [175,176]. The enzyme HMGCR, a key enzyme for cholesterol biosynthesis, upregulated in gastric cancer, activates the Hh/Gli1 signaling pathway. Targeting Hh signaling with statins in combination with cyclopamine has effectively suppressed gastric cancer progression [177].

Inhibition of ACYL in CSCs has not shown significant effects due to the compensatory role of ACSS2, which replenishes acetyl-CoA in the absence of ACLY in cancer models [178]. In vitro inhibition of ACYL in A2780/CDDP ovarian cancer (OC) cells reduced cisplatin resistance by attenuating the PI3K-AKT pathway and activating the AMPK-ROS signaling. These findings suggest that combination of an ACYL inhibitor with cisplatin might represent a promising therapeutic strategy for overcoming cisplatin resistance in OC [179]. Additionally, dephosphorylation and inactivation by PI3K inhibitors have not yielded substantial results in lung cancer treatment [180]. Conversely, inhibiting SCD1 in CSCs has shown promise in impairing cell proliferation, particularly in normal human fibroblasts [181]. The SCD1 inhibitor MK-8245 has demonstrated significant effects in Phase II clinical trials in liver disease [182]. Betulinic acid (BetA) has been identified as a potential inhibitor of SCD in CRC cell lines such as HCT116, SW480, and DLD-1, exhibiting time-dependent antiproliferative effects. In vivo, BetA suppressed tumor growth in a HCT-116 xenograft tumor mouse model [183]. Additionally, BetA induced rapid cell death in CSCs by eliminating their clonogenic capacity [184]. Elevated SCD1 levels, associated with increased MUFAs, have been observed in various cancers, such as lung, ovarian, breast, and GSCs [185]. SCD1 has been shown to regulate the Wnt signaling in CSCs and play a crucial role in CSC maintenance in various cancers, including HCC and CC [15,186,187]. In glioblastoma, Kloosterman et al. demonstrated that lipid-laden macrophages (LLMs), a type of brain-resident microglia that scavenge myelin-derived lipid debris, promote tumor aggressiveness [188]. Inhibiting lipid uptake with the CD36 inhibitor sulfosuccinimidyloleate (SSO) or ABCA1/LXR inhibitors reduced LLM-induced proliferation in mouse glioblastoma models [188,189]. These findings suggest that targeting lipid uptake in CSCs may be a promising therapeutic approach to reduce stemness across various cancers.

Autophagy plays a crucial role in cancer progression and metastasis, as indicated by the elevated autophagy markers expression in CSCs across various cancers. including gastric, colorectal, ovarian, and liver [190]. Numerous small-molecule inhibitors have been developed to target different stages of the autophagy. For example, ULK1, a key regulator of the initiation phase of autophagy, has emerged as a promising target in multiple cancers. Specific inhibitors, such as ULK-101, have demonstrated efficacy in inhibiting autophagy in KRAS-driven lung cancer cells under nutrient deprivation [191]. Other ULK1 inhibitors, such as MRT67307, MRT68921, and SBI-0206965, have shown potential in preclinical studies (Table 2) [192,193]. Inhibitors targeting Vps34, such as 3-MA, wortmannin, LY294002, spautin-1, SAR405, autophinib, vps34-IN1, and PIK-III, require further optimization for dosage and specificity across various cancers [194]. ATG4B, which is highly expressed in cancer and contributes to metastasis and drug resistance, is considered a potential anticancer target. Inhibitors such as Z-FA-FMK, FMK-9a, NSC185058, and S130 have shown significant inhibition of ATG4B, with promising results in preclinical cancer models (Table 2). Late-stage autophagic inhibitors like CQ and HCQ have shown notable effects on cancer cells in vitro and in vivo. New derivatives, such as DC661, have demonstrated significant inhibition of autophagy in specific cancer cell lines, even at low concentrations [195]. Palmitoyl-protein thioesterase 1 (PPT1), a molecular target of HCQ, Lys05, and DC661, is overexpressed in various cancers and is associated with poor prognosis [196]. DC661 has also enhanced the sensitivity of HCC cells to sorafenib [196].

Additionally, inhibitors targeting V-ATPase, which is essential for autophagy and lysosomal acidification, have shown potential in enhancing the anticancer effects of various compounds [197]. For instance, bafilomycin A1 (BafA1) has been found to enhance the efficacy of tyrosine kinase inhibitor (TKI), such as imatinib mesylate, in various cancers [198]. Natural compounds like toosendanin (TSN) and ginsenoside Ro have also demonstrated the ability to inhibit V-ATPase. Notably, ginsenoside Ro has sensitized chemoresistant esophageal cancer cells to 5-fluorouracil (5FU) [199]. Although further investigation is required to elucidate the mechanisms and specificity to these inhibitors across various cancers, their promising effects in combination with anticancer drugs warrant additional validation for potential for clinical applications.

**Table 2 ijms-25-11185-t002:** List of Small Molecules, Cancer Drugs, Autophagy, and Lipid Metabolism Inhibitors in CSCs.

Chemicals, Inhibitors and Drugs	Targeted Molecules and Pathways	Molecular Mechanism	Cancer Types	References
A-922500	Inhibits DGAT1	Reduces LDs Increases cancer cell death	Prostate cancer	[200]
Avasimibe, Avasimin	Inhibits Acyl-CoA cholesterol acyltransferase 1	Increase apoptosis	Colon, prostate, lung, and pancreatic cancer	[201]
AZ22, AZ65	FASN inhibitor	Inhibit FASN-mediated cell growth in cancer	Breast cancer (reduces tumor growth)	[180]
Benzothiazoles and oxalomides, A-939572, CAY10566, CVT-11127	Targets stearoyl-CoA desaturase (SCD)	Inhibits SCD and induces cell cycle arrest	NSCLC, colon, thyroid, and glioblastoma	[202]
BMS309403 (Biphenyl azole compound), BD62694	Inhibits fatty acid binding protein 3 and 4 (FABP3, FABP4)	Reduces lipid accumulation Reduces cell cycle genes (CycD1, VEGFA, and VEGFR)	Prostate cancer, colon cancer, ovarian cancer, and lung metastasis	[203,204]
Cerulenin, C75, C93	Targets FASN targeting CPT1	Reduces stemness markers SOX2, CD133, and FABP7 Induces apoptosis	Glioma stem cells, oesophageal, and squamous cell carcinoma	[205,206]
CAY 10566, SC-26196	Represses NF-κβ activation Promotes AMPK activity and lipophagy	Inhibits SCD1 activity	Induces hepatic steatosis in HSCs	[207]
Cerivastatin	Hydroxy-methylglutaryl-CoA reductase	Inhibits the mevalonate pathway	Breast tumors	[208]
Carbamazepine	Effects on KRAS mutant Prevent steatosis	Effective on early-stage autophagy via ULK1	HCC and colorectal cancer	[209]
Clomipramine, Chloroquine	Induces the expression of PUMA	Defective mitochondrial function	Breast cancer	[194]
Cabozantinib	Inhibits multi-tyrosine kinase	Inhibits VEGF, MET, and AXL	Metastasis renal cell carcinoma, metastatic medullary thyroid cancer, and HCC	[210]
Crizotinib	Inhibits multi-tyrosine kinase	Targets ROS1, EML4, and ALK gene alterations	Non-small cell lung cancer (NSCLC), renal cell carcinoma, HCC, and thyroid cancer	[211,212]
Disulfiram and gemcitabine, etomoxir	Targets aldehyde dehydrogenase ALDH+ breast cancer stem cells	Enhances T-cell immunity Promotes tumorigenesis and drug resistance	Breast cancer and lung cancer	[64,172]
Dacarbazine (DTIC)	Effect on diet-induced obesity and DNA repair	Hyperthermia potentiated its effect in melanoma cell lines	Tumor-bearing HFD-fed mice and Hodgkin lymphoma	[213]
Etomoxir, Perhexiline, ST1326	Inhibits carnitine palmitoyl transferase 1 (CPT1)	Reduces ATP level; Decreases cell viability, inducing apoptosis	BCSC, HCC, and colorectal cancer	[40,90]
Fresolimumab	Blocks the activity of TGF-β isoforms	Shows tumor suppressor response	Breast cancer and renal cell carcinoma	[173,214]
GW3965, LXR623	Liver X receptor (LXR) agonist Agonist of LDR receptors Induced ABC reporter (ABCA1)	Reduces intracellular cholesterol effects on proliferation and tumorigenesis	Clear cell renal cell carcinoma	[215]
GSK165	Inhibits ACLY activity in concentration-dependent manner	Antiproliferative effect on HT29 CRC cells	In CRC-induced sensitivity to anti-neoplastic drug SN38	[216]
Lipofermata and arylpiperazine 5K (DS22420314)	Inhibits the FATP1	Inhibits the uptake of long-chain fatty acids	In ER+ breast cancer	[217]
Metformin and statins	Targets lipid synthesis HMGCR via AMPK-mTOR	Poor prognosis with elevated PLIN1 and DGAT1	Prostate cancer	[200]
2-Methylthio-1,4-naphthoquinone (MTN)	Suppresses lipid uptake and reduces the CSC population	Inhibits CD36 ligands Induced apoptosis in CSC via Caspase 3/7 levels	Glioblastoma multiforme	[218]
ND-646, ND-630, 5-Tetracepoxy-2-furan Acid (TOFA)	Targets ACC1/2	Inhibits FAS Activates FAO	NSCLC lung cancer and HCC	[219]
Orlistat, TVB-2640, GSK2194069	Inhibits FASN activity	Inhibits lipase	Breast, colon, and ovarian cancer	[220,221]
Oligomycin A, antimycin A	Inhibits autophagy	Reduces CSC numbers Induces cytotoxicity	Glioblastoma stem cells	[222]
Resveratrol (3,5,4′-trihydroxystilbene)	Inhibits FASN	Binds to ketoacyl reductase domain	CRC cell proliferation and elevates the apoptosis	[223]
Salinomycin	Acts as ionophore Induces ROS and apoptosis Inhibits lysosomal activity and autophagic flux	Inhibits p-glycoprotein efflux pump Induces apoptosis via reducing CDKN1A/p21 level	ALDH+ cancer cells and BCSCs	[224]
Soraphen A, ND654, 5-tetracepoxy-2-furan acid (TOFA)	Inhibits Acetyl-CoA carboxylase and SCD1	Inhibits ACC catalytic activity	Breast cancer, lung cancer, and prostate cancer	[61,168,169]
Statins, fluvastatin, lovastatin	Inhibits farnesyl pyrophosphate (FPP) and geranylgeranyl pyrophosphate (GGPP) HMG-CoA reductase	Reduces cancer cell proliferation and migration	Prostate, lung, and ovarian cancer	[61]
SSI-4, MF-438, betulinic acid (BetA)	Inhibits Stearoyl-CoA desaturase 1 (SCD)	Induces apoptosis in cancer cells via modulating mitochondrial dynamics, ER stress	HCC, colorectal cancer, and lung cancer	[54]
Simvastatin, LY2157299 (galunisertib)	Inhibits HMG-CoA reductase and EMT antagonist via TGF-β pathway	Reduces vimentin level, β-catenin Inhibits migration and invasion	Bladder cancer, glioblastoma, rectal cancer, pancreatic cancer, HCC, and lung cancer	[13,225]
Sulfosuccinimidyl oleate (SSO)	Targets CD36	Inhibits cancer stem cell growth Reduces migration of CCs	Ovarian cancer and HCC	[226]
5-(tetradecyloxy)-2-furancarboxylic acid (TOFA)	Inhibits ACC activity	Induces apoptosis in dose-dependent manner	In CRC cells HCT-8 and HCT-15	[227]
Vatalanib	Multi-targeted tyrosine kinase, an agonist for VEGFR, PDGFR, and cKit	Drug-resistant cancer cells	Colon cancer	[228]
VY-3-135, Rosiglitazone, 1-(2,3-di (thophen-2-yl) quinoxaline-6-yl)-3-(2-methoxyethyl) urea	Inhibits ACSS2 and ACSL4 activity	PPAR-γ agonist Enhances sensitivity to chemotherapeutic drugs	Breast and bladder cancer	[229]
Vorinostat	Inhibits histone deacetylase	Induces apoptosis and autophagy	Lymphomas, leukemia, and solid tumors	[230]

## 10. Discussion

CSCs pose a significant challenge in various cancer treatments, including chemotherapy, radiotherapy, and immunotherapy, due to their drug resistance. Metabolic reprogramming and autophagy play crucial roles in CSC adaptive responses, making them promising targets to overcome treatment resistance. Notably, targeting lipid metabolism and autophagy may provide therapeutic avenues for eradicating CSCs and enhancing their sensitivity to conventional treatments. CSCs exhibit elevated lipid storage compared to normal cells, promoting resistance to cancer therapy. This lipid accumulation induces FAO and modulates autophagy (Figure 5). Inhibition of key lipid metabolism enzymes and autophagy inhibitors enhances the effectiveness of cancer therapies. For instance, inhibiting FASN sensitizes breast cancer cells to doxorubicin by disrupting lipid synthesis [231]. Similarly, targeting fatty acyl-CoA synthetase (FCS) increases colon cancer cell sensitivity to 5-FU and oxaliplatin by preventing LD accumulation [232]. Moreover, inhibiting glucosylceramide synthase (GCS) in combination with chemotherapy agents, such as paclitaxel (PTX), hydroxyprogesterone (HPR), and irinotecan, has demonstrated synergistic anticancer effects [233]. Targeting fatty acid transport protein 2 (FATP2) in myeloid-derived suppressor cells (MDSCs) reduces fatty acid accumulation, enhances mitochondrial function, and reduces ROS activation, transforming MDSCs into an immune-stimulatory phenotype. This modulation enhances the efficacy of anti-PD-l cancer immunotherapy [234,235]. Additionally, inhibiting CPT1A, a key enzyme in FAO, improves antigen-specific CD8+ T-cell responses and dendritic cell (DC)-mediated T-cell priming, enhancing anti-PD-1 therapy in BRAF^V600E^ melanoma [236]. Combining lipid metabolism modulator with anti-PD-1 therapy and mRNA cancer vaccines could represent a novel and effective regimen for melanoma treatment.

Autophagy, a crucial process in CSC maintenance and survival, is tightly regulated by key molecules. Phosphorylation of ULK1 initiates autophagy by forming a complex with ATG13, FIP200, and other proteins [237,238]. Inhibiting ULK1 has shown promise in sensitizing CSCs to chemotherapy, as evidenced by the ULK1 inhibitor SBI-0206965, which, in combination with mTOR inhibitor, increases cisplatin sensitivity in non-small cell lung cancer [239]. Resveratrol, a natural compound, also inhibits mTORC1 and ULK1, preventing LC3 accumulation and inducing autophagy via AMPK signaling [240]. Deletion of autophagy-related proteins, such as ATG4A, LC3B, and ATG12, has been associated with reduced cancer cell populations and decreased expression of vimentin, a promising marker of cancer progression [156].

Several signaling pathways, including Notch in gastric cancer, JNK/STAT3 in hepatocytes, and EGFR in oral cancer, regulate autophagy and support CSC survival (Figure 5). This highlights autophagy inhibition as a potential therapeutic approach to eliminate CSCs. Inhibitors, such as CQ and HCQ, are under investigation for their ability to reduce autophagy and CSC proliferation in HCC and colorectal cancer. Additional inhibitors, including BafA1 (vacuolar H^+^ ATPase inhibitor), concanamycin A, dimeric-quinapine (DQ661), Lys05 (CQ analog), protease inhibitor E64d, V-ATPase inhibitor gastrin A, 3-methyladenine, and GNS561 (palmitoyl protein thioesterase-1) [241], along with ammonium chloride, methylamine, and siramesine (sigma-2 receptor ligand) [242,243], are undergoing clinical evaluation to determine their efficacy and optimal dosing.

Given the context-dependent nature of autophagy in cancer, which is influenced by tumor type, stage, microenvironment, and genetic factors, a comprehensive evaluation of combination drug approaches is essential. Tailoring therapeutic strategies based on cancer-specific characteristics will aid in the eradication of CSCs, inhibition of their proliferation, and improvement of treatment responses, including chemotherapy, radiotherapy, and immunotherapy.

## 11. Conclusions and Future Perspectives

CSCs are key drivers of recurrence and treatment resistance, sustained by factors like the TME, autophagy, and immune interactions. Targeting CSCs is challenging but crucial for effective cancer eradication. Understanding the mechanisms that assist CSCs, such as quiescent stage, surface markers, CD 133, CD44v6, MET factor receptors (c-MET), epithelial cell adhesion molecules (EpCAM, CD47), and metabolic dependencies, is crucial for developing successful long-term treatments [210]. Metabolic reprogramming, especially targeting lipid and fatty acid metabolism, shows promise in overcoming CSC-mediated resistance [244]. Inhibiting key modulators involved in lipid metabolism and autophagy, such as PNPLA2, FOXO1, LIPA, ULK1, and Vps34, can enhance chemotherapy by increasing CSC sensitivity to cancer therapies (Figure 6).

This review emphasizes targeting lipid metabolism and lipophagy in CSCs. Understanding CSC metabolic dependencies, including FAO, lipogenesis, and LD accumulation, could reveal novel therapeutic targets to overcome drug resistance and prevent tumor relapse. Furthermore, combinations of chemotherapeutic drugs with inhibitors of lipogenesis, lipid uptake, FAO, and autophagy may improve cancer therapy and influence clinical trial outcomes.

## 12. Limitations and Challenges

Targeting lipid metabolism in CSCs is crucial for successful treatment. Not all the CSCs rely on lipid metabolism due to highly heterogenous and often shift to alternative energy sources like glucose, OXPHOS, FAO, and amino acids. This metabolic flexibility makes targeting lipid metabolism alone insufficient for eradicating CSCs. Additionally, multiple lipid pathways are interconnected, so targeting a single pathway may have limited efficacy, highlighting the potential need for combination therapies, which require further research. The risk of off-target effects on normal stem cells and non-cancerous tissues complicates treatment strategies. Thus, identifying key targets in lipid metabolism in CSCs, with minimum off-target effects, remains a critical hurdle.

Furthermore, the enhanced expression of ABC transporters in CSCs leads to drug efflux, reducing the effectiveness of lipid-targeting therapies. The variability in lipid sources, including FAO, de novo lipogenesis, and cholesterol metabolism, across different cancer types adds complexity. In summary, while targeting CSC lipid metabolism holds potential, addressing CSC heterogeneity, metabolic plasticity, and minimizing toxicity to normal cells are major challenges. A deeper understanding of the interplay between CSC lipid metabolism and other metabolic pathways will be crucial for overcoming these obstacles and improving therapeutic outcomes.

## Figures and Tables

**Figure 1 ijms-25-11185-f001:**
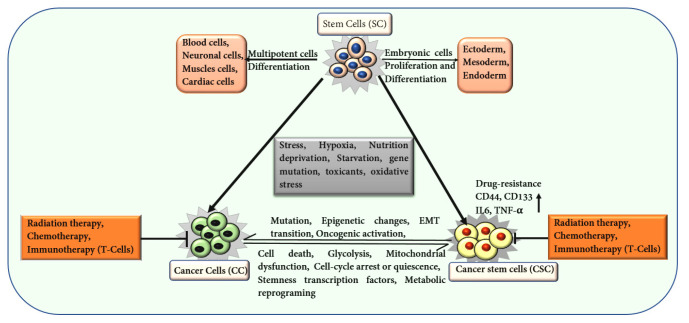
Illustration depicting the characteristics of stem cells (SCs), cancer cells (CCs), and CSCs. The transformation of SCs into CCs and CSCs is influenced by various factors and conditions, which subsequently support cancer cell self-renewal and tumor recurrence. This image highlights associations of CCs and CSCs with various processes, such as stress, chemo and radiation therapy, and metabolic reprogramming, which contribute to CC progression, CSC self-renewal, and metastasis.

**Figure 2 ijms-25-11185-f002:**
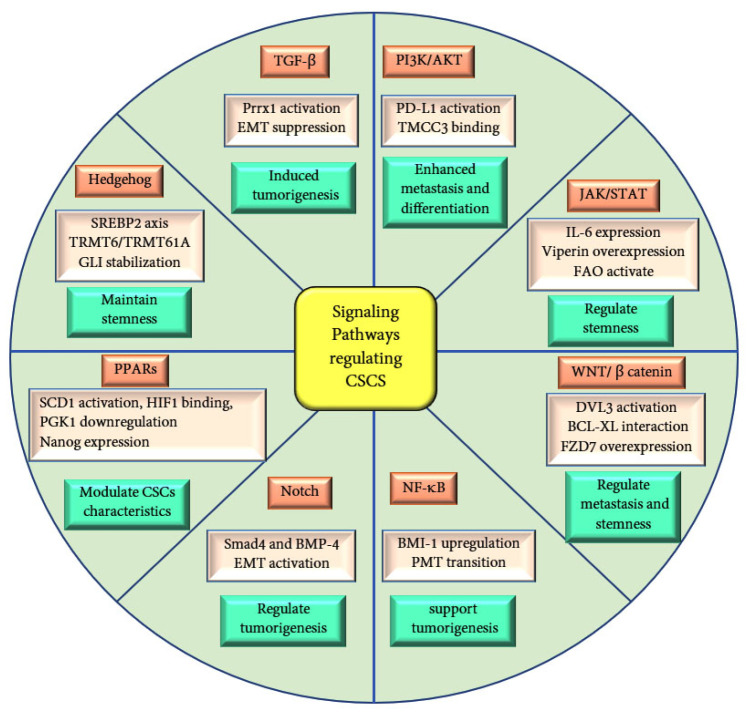
Schematic representation of the key signaling pathways involved in regulating CSCs. The figure highlights critical factors within each pathway that govern CSC characteristics, including stemness, proliferation, survival, and drug resistance. These pathways play a pivotal role in maintaining CSC functions and contribute to cancer progression and therapeutic resistance.

**Figure 3 ijms-25-11185-f003:**
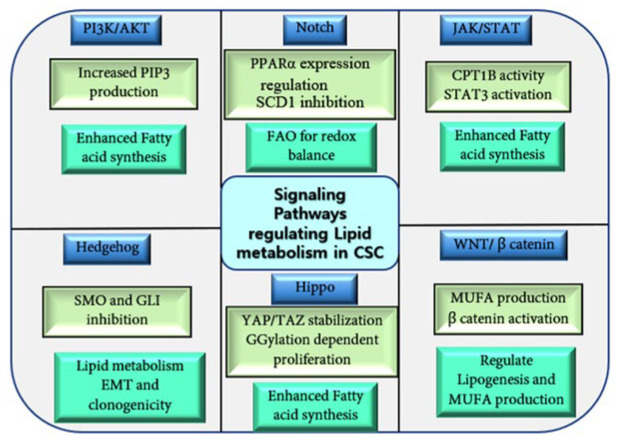
Schematic representation of the key signaling pathways involved in regulating lipid metabolism in CSCs. The figure highlights critical factors within each pathway that regulate lipid metabolism, including FAO, lipogenesis, and LD formation in cancer cells. These pathways are pivotal in controlling lipid levels, their accumulation, and metabolism, all of which contribute to cancer progression and drug resistance.

**Figure 4 ijms-25-11185-f004:**
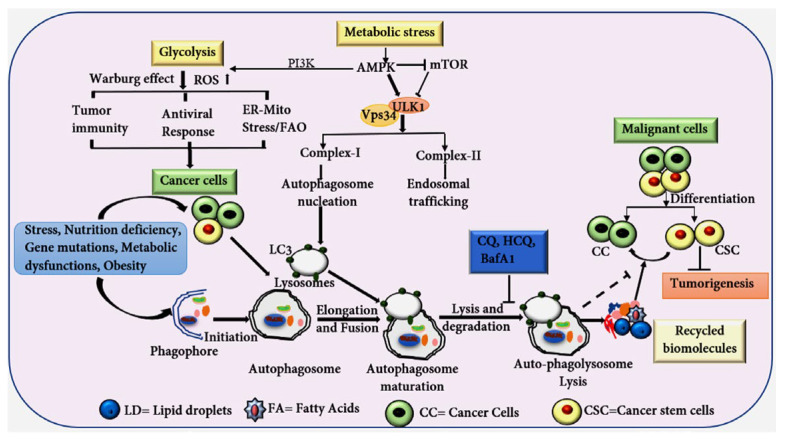
The diagram shows the process of autophagy and its critical role in regulating the stemness of CSC under various stress conditions, such as hypoxia, nutrient deprivation, increased FAO, elevated ROS, therapeutic interventions, and autophagy activation. Autophagy facilitates the removal of damaged organelles, like mitochondria, protein aggregates, and misfolded proteins and lipids. The degradation products are recycled to provide essential nutrients, which support both CCs and CSCs, preserving their stemness. Inhibition of autophagy at either early or late stages significantly impacts the CSC fate, potentially impairing their stemness and survival. Glycolysis and AMPK signaling also support CSC stemness under stress conditions. These pathways support tumor microenvironment (TME), promoting resilience and therapeutic resistance in CSCs.

**Figure 5 ijms-25-11185-f005:**
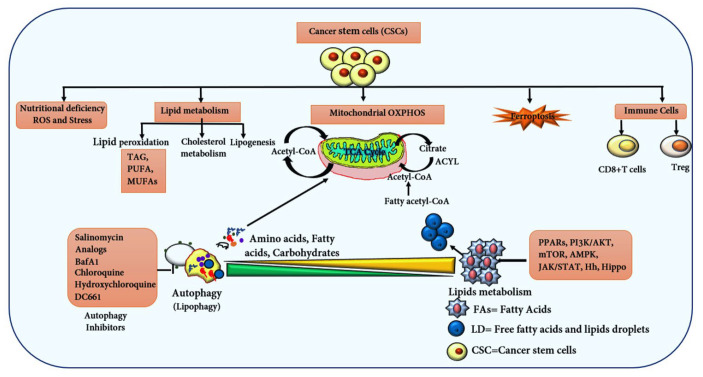
The diagram illustrates the connection between autophagy and lipid metabolism in CSCs, highlighting the central role of mitochondria in sustaining CSCs through metabolites from fatty acid metabolism and autophagy degradation. Key enzymes facilitate the production of Acetyl CoA, crucial for FAO, the TCA cycle, and ATP generation. The diagram also depicts lipid peroxidation and the role of GPX4 in inducing ferroptosis in CSCs. Additionally, CD8+ and Treg cells are shown to regulate CSC stemness. Various inhibitors targeting autophagy, lipid metabolism, and FAO at critical regulatory points are included, aiming to eliminate CSCs.

**Figure 6 ijms-25-11185-f006:**
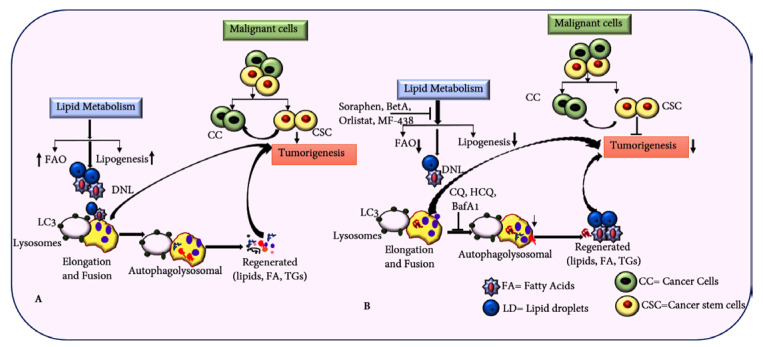
Integrated strategy targeting lipid metabolism and autophagy pathways to eliminate CSCs. (**A**) Illustration depicting how upregulated lipid metabolism supports CSC growth and proliferation. (**B**) Targeting lipid metabolism and autophagy pathways in CC can potentially inhibit tumorigenesis and metastasis driven by CSCs.

## Data Availability

All the references are cited in the manuscript; however, we apologize for the omission of any primary citations.

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
