# Peer review of "Targeting Lipid Metabolism in Cancer Stem Cells for Anticancer Treatment"

_ijms, 2024, doi:10.3390/ijms252011185_

Round 1

Reviewer 1 Report

Comments and Suggestions for Authors

While the manuscript provides detailed mechanisms and in vitro studies, it lacks substantial in vivo or clinical validation. Many of the therapeutic approaches discussed (e.g., targeting SCD1, CPT1A, or autophagy inhibitors) need further preclinical or clinical trial data to establish their efficacy.

The paper presents numerous hypotheses about lipid metabolism pathways and their role in cancer treatment without sufficient experimental backing. This creates a gap between theory and application, which could hinder its impact in the scientific community.

Some of the pathways discussed, such as PI3K/AKT/mTOR and WNT/β-Catenin, have been extensively studied in the context of CSCs. The novelty here primarily stems from integrating lipid metabolism into these pathways, but a significant portion of the manuscript reiterates well-established concepts without adding new insights.

The focus on FASN, CPT1A, and SCD1 as key targets is well-established, but the manuscript lacks alternative novel targets. Broadening the scope to newer, less-explored targets within lipid metabolism may strengthen the study's contribution.

Cancer stem cells exhibit different behaviors depending on the cancer type, and lipid metabolism might not be equally important in all cancers. The manuscript generalizes CSC characteristics, which could be misleading in cancers where lipid metabolism is not a primary factor in stemness.

Comments on the Quality of English Language

Minor editing required

Author Response

Reviewer 1: Comments and Suggestions for Authors

While the manuscript provides detailed mechanisms and in vitro studies, it lacks substantial in vivo or clinical validation. Many of the therapeutic approaches discussed (e.g., targeting SCD1, CPT1A, or autophagy inhibitors) need further preclinical or clinical trial data to establish their efficacy.

Response: We are very thankful and appreciate your insightful comments on our review. The lipid metabolism in CSCs are less explored area and need more insight into different TME conditions. To address your comments, we have incorporated the relevant in vivo and clinical data into the main text. Please go through the revised sections in lines 212-219, 297-304, 383-385, 389, 392-395, 412-413, 526-530, 564-566, and 692-693 indicated in blue color. We have also made efforts to include more recent studies to enhance the significance of recent findings and provide updates on ongoing clinical trials. We are again very thankful to you and believe that the incorporated text helped to improve the quality of our manuscript.

The paper presents numerous hypotheses about lipid metabolism pathways and their role in cancer treatment without sufficient experimental backing. This creates a gap between theory and application, which could hinder its impact in the scientific community.

 Response: We appreciate your valuable comments on our review. We have carefully incorporated data from both in vitro and in vivo cancer models, as well as patient studies. The revised text can be found in lines 95-99, 110-112, 189-192, 197-200, 214-219, 286-290, 306-307, 326-331, 372-376, 395-399, 408-419, 526-531, and 553-557. These updates include background studies and their findings, which help bridge the gap between theoretical knowledge and practical application. Additionally, we have provided more detailed explanations to enhance the review's significance and impact within the scientific community.

Some of the pathways discussed, such as PI3K/AKT/mTOR and WNT/β-Catenin, have been extensively studied in the context of CSCs. The novelty here primarily stems from integrating lipid metabolism into these pathways, but a significant portion of the manuscript reiterates well-established concepts without adding new insights.

 Response: We have carefully considered your opinion on the current review. We have addressed the points raised by discussing various signaling pathways and key regulatory enzymes that could be effective targets for CSCs in different cancers. For example, in line 540, we mentioned that blocking TGF-β targets the immune pathway and impairs breast cancer stem cells (BCSCs) in vivo. Additionally, targeting ACC, which is involved in palmitoylation, is highlighted as a novel approach in cancers dependent on Wnt and Hedgehog (Hh) signaling. In line 569, we discuss how SCD1's regulation of Wnt signaling presents a promising target for hepatocellular carcinoma (HCC) and cholangiocarcinoma (CC), with ongoing preclinical trials. We have incorporated many key regulatory molecules that have potential as novel therapeutic targets and are currently being explored in clinical trials.

The focus on FASN, CPT1A, and SCD1 as key targets is well-established, but the manuscript lacks alternative novel targets. Broadening the scope to newer, less-explored targets within lipid metabolism may strengthen the study's contribution.

Response: We have addressed the mentioned issues and highlighted several molecules that could serve as novel targets for CSCs in various cancers. Specifically, we have discussed additional targets such as PPARα (lines 87-91), vps34 (line 581), Nanog (lines 95-98), IL-6 (lines 153-157), PDK1 (lines 189-192), ALDH1A1 (lines 193-197), MUFAs and PUFA-Pls (lines 248-254), SERBPs (lines 259-263), SR-B1 (lines 293-294), LXR (lines 300-303), MMD (lines 340-344), FSP1 (line 403), ULK1 (line 575), LDs, and PNPLA2 (line 502), all of which are underexplored in relation to CSCs and may hold potential as therapeutic targets in lipid metabolism for clinical trials.

Additionally, we have included two tables (Table 1 and Table 2) and two more figures that list various enzymes and inhibitors involved in lipid metabolism and CSCs, which could be promising for discovering new therapeutic targets in cancer. We appreciate your constructive feedback, which has helped us improve the clarity and depth of our review.

Cancer stem cells exhibit different behaviors depending on the cancer type, and lipid metabolism might not be equally important in all cancers. The manuscript generalizes CSC characteristics, which could be misleading in cancers where lipid metabolism is not a primary factor in stemness.

Response: Our focus is on lipid metabolism and autophagy-mediated functions in eradicating cancer stem cells (CSCs), a topic that remains incompletely understood. We have discussed the signaling pathways and key regulatory enzymes in CSCs that may serve as potential diagnostic and therapeutic targets in cancers where lipid metabolism plays a crucial role in tumorigenesis. However, we recognize the importance of considering the metabolic vulnerabilities of non-cancerous cells within the tumor microenvironment. Therefore, a precise evaluation of lipid metabolites across different cancer origins is essential to determine when lipid metabolism plays a pivotal role, as opposed to cancers where it is not the primary factor.

Reviewer 2: Comments and Suggestions for Authors

Singh et.al. reviewed the lipid metabolism in cancer stem cells (CSCs). In the beginning, they clearly explained about CSCs and the signaling pathways that regulate stemness and promote tumor progression. They reviewed the lipid metabolism in CSC, its signaling pathways, and the role of autophagy in maintaining lipid levels. Tables and figures are very well presented and help the readers to follow the text. The review is very interesting and easy to follow and understand. One suggestion to improve the review:

  • It will be nice to have a figure to explain the signaling pathways involved in lipid metabolism in CSCs. It will help the reader to follow the session.

Response: In response to your valuable comments on our review. To further address your suggestion, we have incorporated separate figures (Figure 2 and Figure 3) that illustrate the signaling pathways involved in CSCs and lipid metabolism. These additions aim to enhance clarity and make the content more accessible for readers. We appreciate your feedback and believe these revisions will improve the overall understanding of the manuscript.

Reviewer 2 Report

Comments and Suggestions for Authors

Singh et.al. reviewed the lipid metabolism in cancer stem cells (CSCs). In the beginning, they clearly explained about CSCs and the signaling pathways that regulate stemness and promote tumor progression. They reviewed the lipid metabolism in CSC, its signaling pathways, and the role of autophagy in maintaining lipid levels. Tables and figures are very well presented and help the readers to follow the text. The review is very interesting and easy to follow and understand. One suggestion to improve the review:

1-       It will be nice to have a figure to explain the signaling pathways involved in lipid metabolism in CSCs. It will help the reader to follow the session.

Author Response

Reviewer 2: Comments and Suggestions for Authors

Singh et.al. reviewed the lipid metabolism in cancer stem cells (CSCs). In the beginning, they clearly explained about CSCs and the signaling pathways that regulate stemness and promote tumor progression. They reviewed the lipid metabolism in CSC, its signaling pathways, and the role of autophagy in maintaining lipid levels. Tables and figures are very well presented and help the readers to follow the text. The review is very interesting and easy to follow and understand. One suggestion to improve the review:

  • It will be nice to have a figure to explain the signaling pathways involved in lipid metabolism in CSCs. It will help the reader to follow the session.

Response: In response to your valuable comments on our review. To further address your suggestion, we have incorporated separate figures (Figure 2 and Figure 3) that illustrate the signaling pathways involved in CSCs and lipid metabolism. These additions aim to enhance clarity and make the content more accessible for readers. We appreciate your feedback and believe these revisions will improve the overall understanding of the manuscript. 

Round 2

Reviewer 1 Report

Comments and Suggestions for Authors

The Authors have addressed the Major points raised by this reviewer in order to obtain a better manuscript.

Two points remain to be addressed:

1) The following previous comments has been not sufficiently addressed:

Cancer stem cells exhibit different behaviours depending on the cancer type, and lipid metabolism might not be equally important in all cancers. The manuscript generalises CSC characteristics, which could be misleading in cancers where lipid metabolism is not a primary factor in stemness.

The answer of the Authors has been the following:

Response: Our focus is on lipid metabolism and autophagy-mediated functions in eradicating cancer stem cells (CSCs), a topic that remains incompletely understood. We have discussed the signaling pathways and key regulatory enzymes in CSCs that may serve as potential diagnostic and therapeutic targets in cancers where lipid metabolism plays a crucial role in tumorigenesis. However, we recognize the importance of considering the metabolic vulnerabilities of non-cancerous cells within the tumor microenvironment. Therefore, a precise evaluation of lipid metabolites across different cancer origins is essential to determine when lipid metabolism plays a pivotal role, as opposed to cancers where it is not the primary factor. 

The above answer is either insufficient and not a response.

The Authors must highlight this point as a Limitation of the manuscript. When they do this it is needed that they alter their Conclusions (that now are too long), writing in a separate subheading the Limitations with respect to the Conclusions and Perspectives. In this way the Conclusions and Perspectives will be much better focused.

2) The number of citation is excessive. A specific comments has been written separately by myself indication how they must reduce the total number.

Comments on the Quality of English Language

Minor editing is needed.

Author Response

Reviewers comments:

The Authors have addressed the Major points raised by this reviewer in order to obtain a better manuscript.

Two points remain to be addressed:

1) The following previous comments has been not sufficiently addressed:

Cancer stem cells exhibit different behaviors depending on the cancer type, and lipid metabolism might not be equally important in all cancers. The manuscript generalizes CSC characteristics, which could be misleading in cancers where lipid metabolism is not a primary factor in stemness.

The answer of the Authors has been the following:

Response: Our focus is on lipid metabolism and autophagy-mediated functions in eradicating cancer stem cells (CSCs), a topic that remains incompletely understood. We have discussed the signaling pathways and key regulatory enzymes in CSCs that may serve as potential diagnostic and therapeutic targets in cancers where lipid metabolism plays a crucial role in tumorigenesis. However, we recognize the importance of considering the metabolic vulnerabilities of non-cancerous cells within the tumor microenvironment. Therefore, a precise evaluation of lipid metabolites across different cancer origins is essential to determine when lipid metabolism plays a pivotal role, as opposed to cancers where it is not the primary factor. 

The above answer is either insufficient and not a response.

New Response: CSCs share some common characteristics across cancer types, their behavior and metabolic dependencies can vary significantly depending on the tumor microenvironment and origin. While, lipid metabolism is crucial for CSC stemness in some cancer, influencing processes like energy production, membrane synthesis, and signaling pathways. However, not all cancers rely heavily on lipid metabolism. In such cases, other metabolic pathways such as glycolysis, OXPHOS may play a more dominant role in CSC maintenance. Understanding the metabolic landscape of each cancer type is essential for developing more precise therapeutic strategies of targeting CSCs. In such cases, targeting non-lipid pathways may be more effective.

Key alternative strategies to target CSCs in non-lipid dependent cancers include:

  1. Glucose and amino acid metabolism such as glycolysis and OXPHOS, glutamine, glycine, and serine uptake are critical for redox balance and nucleotide synthesis could act as an alternative source of energy. Thus, targeting these pathways could be an effective strategy in those cancers.
  2. Another key regulator is genetic and epigenetic modulation, which maintain undifferentiated and quiescence state of CSCs, thus targeting those genes and proteins may reduce stemness in CSCs.
  3. Non-lipid signaling pathways, such as Wnt, Hedgehog, Notch could be on important player in those cancers.
  4. CSCs maintain low levels of ROS by scavenging mechanism to preserve their stemness. Inhibiting alternative antioxidant mechanism like glutathione and NADPH to induce ROS and reduce CSC survival.
  5. The tumor microenvironment (TME) influences lipid availability. In lipid-rich TMEs, CSCs may rely on external lipid sources, making intracellular lipid-targeting drugs less effective. In hypoxic or nutrient-deprived TMEs, CSCs may prioritize other metabolic pathways.

Limited understanding of specific lipid targets also influences the designing of specific target across various cancers. A deeper insight in CSCs lipid metabolism and its interactions with other cellular pathways would be crucial to overcome these limitations.

The Authors must highlight this point as a Limitation of the manuscript. When they do this it is needed that they alter their Conclusions (that now are too long), writing in a separate subheading the Limitations with respect to the Conclusions and Perspectives. In this way the Conclusions and Perspectives will be much better focused.

Response: In response to your insightful feedback. We have addressed your suggestion by incorporating a dedicated sub-heading on limitations and challenges. Please refer to lines 679-696 in the revised text. We appreciate your valuable input and believe on these revisions that enhance the clarity and depth of the manuscript.

2) The number of citations is excessive. A specific comment has been written separately by myself indication how they must reduce the total number.

Response: We have incorporated the relevant references throughout the text in the review. We would greatly appreciate your guidance on how to streamline the number of references. Your advice will help us adopt a more effective approach in future manuscripts.